# Analyzing the impact of socioeconomic indicators on gender inequality in Sri Lanka: A machine learning-based approach

Sherin Kularathne[1], Amanda Perera[2], Namal Rathnayake [3]*, Upaka Rathnayake [4], Yukinobu Hoshino [5]

1 Faculty of Graduate Studies and Research, Sri Lanka Institute of Information Technology, Malabe, Sri Lanka, 2 Department of Business Economics, Faculty of Management Studies and Commerce, University of Sri Jayewardenepura, Gangodawila, Nugegoda, Sri Lanka, 3 River and Environmental Engineering Laboratory, Graduate School of Engineering, The University of Tokyo, Bunkyo City, Tokyo, Japan, 4 Department of Civil Engineering and Construction, Faculty of Engineering and Design, Atlantic Technological University, Sligo, Ireland, 5 School of Systems Engineering, Kochi University of Technology, Kami, Kochi, Japan

* namal@hydra.t.u-tokyo.ac.jp

**Data Availability Statement:** All data files are available from the Kaggle database named as Sri Lankan Economic Indicators Dataset (SLEID) (https://kaggle.com/datasets/8b7511e075859

## Abstract

This study conducts a comprehensive analysis of gender inequality in Sri Lanka, focusing on the relationship between key socioeconomic factors and the Gender Inequality Index (GII) from 1990 to 2022. By applying machine learning techniques, including Decision Trees and Ensemble methods, the study investigates the influence of economic indicators such as GDP per capita, government expenditure, government revenue, and unemployment rates on gender disparities. The analysis reveals that higher GDP and government revenues are associated with reduced gender inequality, while greater unemployment rates exacerbate disparities. Explainable AI techniques (SHAP) further highlight the critical role of government policies and economic development in shaping gender equality. These findings offer specific insights for policymakers to design targeted interventions aimed at reducing gender gaps in Sri Lanka, particularly by prioritizing economic growth and inclusive public spending.

## Introduction

In a world striving for gender equality, disparities persist, impacting women's socio-economic standing and opportunities for advancement [1]. Globally, gender inequality remains a pressing concern, with women often facing limited access to education, healthcare, and economic resources compared to their male counterparts [2]. With less than a decade remaining to fulfil the 2030 Agenda for Sustainable Development, it is critical to ensure that everyone has equal access to education, healthcare, decent work, a life free of discrimination and abuse, and representation in political and economic decision-making. It is a necessary foundation for a peaceful and sustainable planet, as well as for the development of a prosperous society [3]. According to World Bank data shown in Fig 1, women will make up 52% of Sri Lanka's population in 2022, contributing to a demographic trend that highlights a significant and

a13545b4503da97749d3b7b5c61277b934537efa
4fdc26563aa).

**Funding:** This study was funded by the Japan
Society for the Promotion of Science (JSPS) in the
form of a Grants-in-Aid for Scientific Research
(KAKENHI) grant to Prof Yukinobu Hoshino. Grant
Number: 22KK0160. The funders were involved in
supervision, decision to publish, and verification of
the study.

**Competing interests:** The authors have declared
that no competing interests exist.

continuous 4.6% increase in the percentage of the female population over the course of more than six decades. This is in stark contrast to the global trend, which has seen a 0.26% reduction over the same time period [4]. As a result, promoting gender equality is critical to unlocking the potential of Sri Lanka's expanding female population.

In Sri Lanka, mainly due to the political instability in recent decades, gender disparities persist, particularly in areas such as employment, political representation, and access to decision-making roles [5]. Providing evidence for the above, the female population only represents 5.3% of the parliament, which is an indication of severe underrepresentation, and women's labour force participation is 33.6% as of 2021, underscoring ongoing difficulties in reaching gender parity in the workforce. The United Nations Population Fund executed research in 2019 that revealed a startling 90% of Sri Lankan women and girls had experienced sexual harassment in public transport [6]. This finding highlights the widespread prevalence of gender-based violence and harassment in public spaces. These concerning figures highlight the urgent requirement for thorough studies that not only pinpoint gender discrepancies in Sri Lanka but also examine the structural barriers and cultural norms that support them [3]. These disparities not only hinder women's advancement but also impede overall economic growth and development. Understanding the factors contributing to gender inequality and their implications is crucial for policymakers and stakeholders seeking to promote inclusive and sustainable development.

In previous studies on gender inequality in Sri Lanka, there has been a notable gap in comprehensive analyses that simultaneously consider multiple economic indicators and their impact on the Gender Inequality Index (GII). While some studies have focused on specific aspects of gender inequality, such as education or workforce participation, few have undertaken a holistic approach that considers the broader socioeconomic context. For example, a study by Seneviratne [7] highlighted the need for a more nuanced understanding of gender disparities in Sri Lanka, pointing to the limited scope of existing research in capturing the complex interplay of economic factors that contribute to gender inequality. This gap in the literature underscores the importance of conducting a comprehensive analysis that considers a wide range of economic indicators and their implications for gender inequality in Sri Lanka.

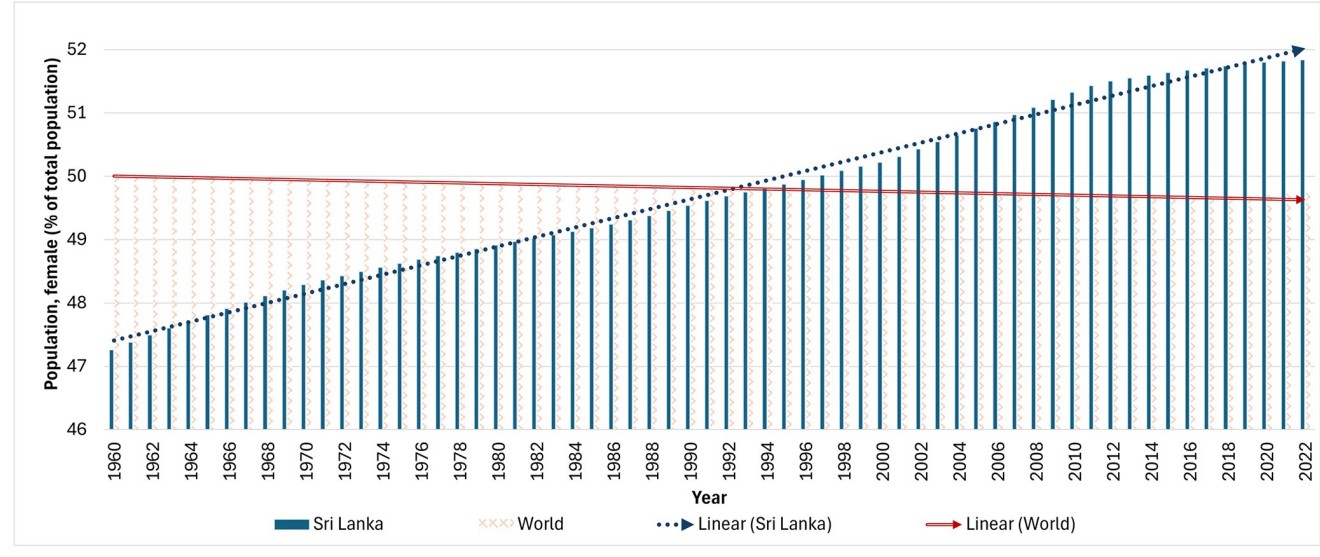

**Fig 1. Female population growth trend comparison.**

To address this gap, our study aims to provide a holistic analysis of gender inequality in Sri Lanka by examining the relationship between various economic indicators and the GII. Building on the work of previous studies, such as that of Gunawardena et al. [8], which focused on gender disparities in education, our research seeks to expand the scope of analysis to include a broader range of economic factors. By employing machine learning (ML) algorithms to analyze a comprehensive dataset spanning multiple economic indicators, we aim to provide a more nuanced understanding of the drivers of gender inequality in Sri Lanka. This approach is expected to contribute significantly to the existing literature by offering new insights into the complex relationship between economic indicators and gender inequality in the country.

## Literature review

### Gender inequality in developing countries, South Asia, and Sri Lanka

Gender inequality remains a significant issue, as evidenced by its inclusion as one of the United Nations' Sustainable Development Goals. However, progress toward achieving gender equality is falling short, with the goal becoming increasingly elusive. If present trends persist, over 340 million women and girls will continue to live in extreme poverty by 2030 [9]. The GII is the primary measure of gender inequality, encompassing three main dimensions: reproductive health, empowerment, and the labor market. The maternal mortality ratio and adolescent birth rate are key factors within the reproductive health dimension. The empowerment dimension focuses on educational attainment and parliamentary representation, while the labor market dimension considers the labor force participation rate [10].

The developing world is considered more vulnerable in achieving gender equality. For instance, Jayachandran [11] argues that societal norms, like patrilocality and concerns over women's "purity," heighten the favouritism toward males. Additionally, gender disparities in outcomes are often most pronounced in low-income countries with the most challenging socio-economic conditions [12]. Despite advancements in gender equality, women in the most disadvantaged low- and middle-income countries continue to face significant barriers. These obstacles, including limited access to resources, impede their life outcomes and those of their children [13].

In South Asia, gender inequality remains notably high, with significant gaps evident in labor force participation, parliamentary representation, and the presence of women in senior managerial roles [14]. Sri Lanka, a developing country in South Asia, also shows significant disparities in labor force participation and parliamentary representation. In 2022, the gender gap was recorded at 41% for labor force participation and 89.3% for parliamentary representation [10].

### Socio economic factors and gender inequality

Numerous studies in this area examine the factors contributing to gender inequality, with a particular emphasis on specific sectors. For example, Amarasooriya [15] explored the obstacles that impede the business growth of women entrepreneurs in small and medium-sized enterprises, while Premarathna [16] investigated the socio-cultural factors contributing to the gender pay gap in Sri Lanka's rural agricultural sector. There is a notable lack of research that integrates a holistic view of socioeconomic factors with advanced ML techniques to derive policy implications. This study attempts to fill this gap by offering a novel approach that leverages ML to decode the complex interactions between socioeconomic factors and gender inequality. This approach allows for the identification of subtle patterns and trends that traditional methods might miss, thereby providing more robust policy recommendations.

The primary objective of this study is to investigate the relationship between key economic indicators and the GII in Sri Lanka using advanced ML algorithms. Building upon existing research [17–19], this study considers the following indicators to investigate the influence of GII. A detailed explanation of each indicator is given in S1 Table.

- GDP based on purchasing power parity, the share of the world (GDS),

- GDP per capita, current prices (GDC),

- GDP, current prices (GDP),

- Government expenditure (GEX),

- Government primary balance, percent of GDP (GPB),

- Government primary expenditure, percent of GDP (GPE),

- Government revenue, percent of GDP (GRV),

- Unemployment rate (UER)

The study aims to contribute to the existing literature by providing a comprehensive analysis that considers multiple economic indicators simultaneously, utilizing state-of-the-art ML techniques.

Based on existing literature, we hypothesise that there is a significant relationship between various economic indicators and GII in Sri Lanka. Specifically, we hypothesise that GDS [20], GDC [21], GDP [22], GEX [23], GPB [24], GPE [25], GRV [26], and UER [20] are all significant predictors of GII.

Following the existing theoretical framework, the following hypothesis can be introduced in this study.

1. Hypothesis 1: GDC and GRV positively impact GII.
   This hypothesis is supported by the Solow-Swan Growth Model [32] and Sen's Capability Approach [33–35], which suggest that economic development creates opportunities for women, improving their social and economic standing. Stotsky's fiscal policy framework [36] further strengthens this, arguing that strategic public investments in women's education and healthcare can directly reduce gender inequality.

2. Hypothesis 2: GEX and GPE reduce gender inequality.
   This can be grounded in Stotsky's Gender-Responsive Fiscal Policy [37], which posits that targeted government spending in social sectors helps close gender gaps, empowering women in both the workforce and politics.

3. Hypothesis 3: Higher UER exacerbate gender inequality.
   This connects to Social Role Theory and Intersectionality Theory [38], suggesting that women are more vulnerable to unemployment due to structural inequalities in labor markets and that these disparities are more pronounced for women from marginalized groups.

4. Hypothesis 4: Higher GPB may negatively impact gender inequality.
   Based on Public Finance Theory [39, 40], poor fiscal management means fewer resources are available for social programs targeting gender gaps, making it harder for women to access health, education, and employment opportunities.

Therefore, We expect that positive economic growth, reflected in higher GDP and government revenue, will lead to lower gender inequality. Conversely, higher levels of unemployment and government expenditure relative to GDP may exacerbate gender disparities. We also

anticipate that previous policies and government actions [43] will have a lasting impact on the current state of gender inequality, which we aim to uncover through our analysis.

The selection of key economic indicators such as GDP, GEX, and the UER as predictors of the GII in Sri Lanka is grounded in established economic and gender theories. The GDP is widely recognized as a fundamental measure of economic performance, with higher GDP per capita often linked to enhanced social development and reduced gender disparities as economic growth typically expands opportunities for both men and women [44]. Government expenditure and revenue, representing fiscal policy tools, are also critical in influencing gender inequality. Theories of public finance suggest that strategic government spending on education, healthcare, and social welfare can directly reduce gender gaps by improving access to services that empower women [36]. Conversely, higher unemployment rates often exacerbate gender disparities, as women are disproportionately affected by job losses due to structural inequalities in the labor market [45]. This study builds upon these theoretical foundations, using advanced ML algorithms to quantify the relationships between these economic indicators and GII, aiming to uncover the nuanced impacts of economic policies and performance on gender inequality in Sri Lanka.

A more detailed explanation of the variable selection is presented in Table 1.

The theoretical basis for this study is based on a variety of economic and social theories, as well as major findings from previous research. The Solow-Swan Growth Model and Sen's Capability Approach propose that economic growth, as measured by GDC and GRV, can improve gender equality by increasing women's access to resources and opportunities. This is corroborated by Forsythe et al. [22] and Birchall and Fontana [41], who discovered that economic expansion improves social development while decreasing gender gaps. According to Stotsky's Fiscal Policy Framework, government revenue policies that promote equitable and gender-sensitive public services have the potential to reduce gender disparity.

In terms of GEX and GPE, Stotsky's Gender-Responsive Fiscal Policy posits that targeted public spending on social sectors like education and healthcare reduces gender gaps by improving women's access to essential services. This is corroborated by Donald and Lusiani [42] and Abramovsky and Selwaness [27], who agrees that social expenditure improves women's chances and minimises unpaid labour.

Social Role Theory and Intersectionality Theory explain the relationship between UER and gender inequality by emphasising the structural impediments that women encounter in the labour market. Faďoš and Bohdalová [28] and Baussola and Mussida [29] found that increased unemployment disproportionately affects women, aggravating gender gaps in employment.

**Table 1. Variables and supporting studies.**

| Hypotheses | Variables | Past Literature | | |
|---|---|---|---|---|
| | | Expected Sign | Theories | Related Past Studies |
| 1 | GDC—>GII | (+) | Solow-Swan Growth Model [32] | [22], |
| | GRV—>GII | | Sen's Capability Approach [33–35] | [41] |
| | | | Stotsky's Fiscal Policy Framework [36] | |
| 2 | GEX—>GII | (-) | Stotsky's Gender-Responsive Fiscal Policy [37] | [42] |
| | GPE—>GII | | | [27] |
| 3 | UER—>GII | (-) | Social Role Theory and Intersectionality Theory [38] | [28] |
| | | | | [29] |
| 4 | GPB—>GII | (+), (-) | Public Finance Theory [39, 40] | [30] |
| | | | | [31] |

Finally, Public Finance Theory explains the GPB's mixed effects on gender inequality. While fiscal responsibility may enhance available resources for social investment, austerity measures can reduce women's access to services. Research by Karamessini and Rubey [30] and Onaran [31] suggests that austerity policies can worsen gender disparity. This theoretical framework and prior research inform the variables and hypotheses explored in this study.

## Machine learning for economics

In recent years, ML has emerged as a powerful tool for predicting and forecasting time series data [46–48]. Studying socioeconomic issues, including gender inequality, showed promising results using ML and AI methods [49, 50]. ML techniques, such as decision trees, random forests, and neural networks, have been used to analyse large datasets and identify patterns and trends in gender disparities [51]. These approaches have the potential to provide valuable insights into the complex interactions between economic indicators and gender inequality, helping policymakers design more effective interventions [52, 53].

Despite the promise of ML in studying gender inequality, there are challenges and limitations to consider. One challenge is the need for high-quality data, particularly in developing countries like Sri Lanka, where data collection systems may be limited. Additionally, ML algorithms are often complex and may require specialized expertise to interpret and apply effectively. Furthermore, there are ethical considerations regarding the use of ML in socioeconomic studies, such as the potential for bias in algorithmic decision-making [54].

Future research in this area should focus on addressing the limitations of existing studies and advancing our understanding of the relationship between economic indicators and gender inequality in Sri Lanka. This could involve the development of more sophisticated ML models that can account for complex interactions and nonlinear relationships. Additionally, research should continue to explore the impact of public policies and interventions on gender disparities, using ML to inform evidence-based policy-making [55].

## Problem formulation

Despite significant progress in reducing gender inequality in Sri Lanka, disparities persist in various aspects of life, including education, employment, and political representation. Existing studies have primarily focused on analyzing the impact of individual economic indicators on gender inequality, often overlooking the complex interactions among these indicators. Moreover, while ML has been increasingly used in socioeconomic studies, its application to gender inequality in Sri Lanka remains limited. Therefore, this study aims to fill this gap by employing ML techniques to comprehensively analyze the relationship between multiple economic indicators and the GII in Sri Lanka. By doing so, this research seeks to provide a deeper understanding of the underlying factors contributing to gender disparities in the country and inform evidence-based policy-making.

The following steps are looked at to fill the research gap.

1. To analyze the relationship between various economic indicators (GDS, GPC, GDP, GEX, GPB, GPE, GRV, UER) and the GII in Sri Lanka using ML algorithms.

2. To compare the performance of different ML algorithms in predicting the GII based on economic indicators, identifying the most effective approach for analyzing gender inequality in Sri Lanka.

3. To identify the impact of each economic indicator when ML model training by using explainable machine learning techniques (SHAP).

4. To discuss the previous policies and government actions on gender inequality in Sri Lanka by examining the historical trends of economic indicators and their correlation with the GII.

## Materials and methodology

### Dataset

Table 2 provides a summary of key economic indicators and the GII in Sri Lanka from 1990 to 2022. The indicators include GDS, GPC, GDP, GEX, GPB, GPE, GRV, and UER. The data shows variations in these indicators over time, with mean values indicating the average level of each indicator across the years and standard deviations reflecting the degree of variability.

The key variables used in this analysis are detailed below, including their definitions, measurement methods, and sources.

1. Gender Inequality Index (GII)
   The GII is the dependent variable in this study. It represents gender-based disparities in three critical dimensions: reproductive health, empowerment, and labor market participation. The GII ranges from 0, indicating perfect gender equality, to 1, reflecting complete gender inequality. This index is a composite measure that captures the extent of inequality between men and women in these areas and is sourced from the United Nations Development Programme (UNDP).

2. Gross Domestic Product (GDP)
   GDP is a key economic indicator used in the study, representing the total value of goods and services produced within a country in a given period, adjusted for inflation (real GDP). It is a measure of overall economic performance and is expressed in current prices. Higher GDP levels are generally associated with improved economic conditions and potentially lower gender inequality. The GDP data used in this study is obtained from the World Bank.

3. GDP Per Capita (GDC)
   GDC is another crucial economic indicator, calculated by dividing the GDP by the total population. This variable provides insight into the average economic output per person, reflecting the standard of living in the country. Higher GDP per capita often correlates with better access to resources and opportunities, which can influence gender inequality. The data for GDP per capita is sourced from IMF.

4. Government Expenditure (GDX)
   GEX refers to the total government spending as a percentage of GDP, including

**Table 2. Summary statistics of key economic indicators and the Gender Inequality Index (GII) in Sri Lanka from 1990 to 2022.** (The data acquisition from the International Monetary Fund (IMF) [56] and World Bank data repositories [57]).

|        | GDS  | GDC   | GDP    | GEX   | GPB   | GPE   | GRV   | UER   | GII  |
|--------|------|-------|--------|-------|-------|-------|-------|-------|------|
| count  | 33   | 33    | 33     | 33    | 33    | 33    | 33    | 33    | 33   |
| mean   | 0.18 | 7624  | 156.42 | 20.57 | -1.78 | 15.54 | 13.76 | 7.82  | 0.42 |
| std    | 0.03 | 4298  | 97.67  | 2.44  | 1.45  | 2.46  | 2.67  | 3.59  | 0.04 |
| min    | 0.13 | 2225  | 38.55  | 16.61 | -5.93 | 11.99 | 8.32  | 4.00  | 0.36 |
| 25%    | 0.16 | 3990  | 73.71  | 19.13 | -2.38 | 13.79 | 12.59 | 4.80  | 0.38 |
| 50%    | 0.17 | 6468  | 127.41 | 20.15 | -1.58 | 15.01 | 13.22 | 6.60  | 0.42 |
| 75%    | 0.22 | 11706 | 243.24 | 21.73 | -0.88 | 16.97 | 14.93 | 9.20  | 0.45 |
| max    | 0.24 | 14621 | 323.95 | 26.42 | 0.60  | 21.58 | 18.96 | 15.90 | 0.47 |

consumption, investment, and transfer payments. This variable is critical for understanding the role of fiscal policy in shaping gender inequality. Higher government expenditure might lead to improved public services, which can reduce gender disparities, but it could also exacerbate inequality if not properly targeted. The GEX data is sourced from the IMF.

5. Government Revenue (GRV)
   GRV is the income generated by the government from taxes and other sources, expressed as a percentage of GDP. It is a key indicator of the government's capacity to finance public services, which can impact gender equality by providing resources for education, healthcare, and social programs. The GRV data used in this study is sourced from the IMF.

6. Government Primary Balance (GPB)
   GPB is the government's fiscal balance, excluding interest payments on outstanding debt, expressed as a percentage of GDP. A positive primary balance indicates that the government is generating more revenue than spending (excluding debt interest), which may provide more room for gender-focused investments. The data for GPB is sourced from the IMF.

7. Government Primary Expenditure (GPE)
   GPE measures the government's spending on goods, services, and transfer payments, excluding interest payments on debt, as a percentage of GDP. This variable helps in understanding the government's spending priorities and its potential impact on gender inequality. The data for GPE is obtained from the IMF.

8. Unemployment Rate (URE)
   UER represents the percentage of the labor force that is unemployed and actively seeking employment. It is a key indicator of labor market health and can have significant implications for gender inequality, particularly in terms of access to economic opportunities. Higher unemployment rates may exacerbate gender disparities, especially if women are disproportionately affected. The UER data used in this study is sourced from the World Bank.

The correlation analysis reveals strong relationships between GII and several key economic indicators in Sri Lanka (Refer to Fig 2). Specifically, there is a strong negative correlation between GII and GDC (-0.944), indicating that as GDC increases, GII tends to decrease. Similarly, GDP shows a strong negative correlation (-0.941), suggesting that overall economic output is inversely related to gender inequality. The UE also exhibits a strong positive correlation with GII (0.771), implying that higher levels of unemployment are associated with greater gender inequality.

These results highlight the robust correlation between gender inequality in Sri Lanka and economic issues. Nevertheless, as this research does not reveal the direction of causality, consideration must be used in interpreting the results. Whether and how economic variables cause gender inequality, or vice versa, should be the subject of future research using experimental or longitudinal methodologies.

Furthermore, the analysis reveals interesting insights into the role of GEX and GRV in gender inequality. While GEX shows a positive correlation (0.691) with GII, indicating that higher government spending relative to GDP is associated with greater gender inequality, GRV exhibits a strong positive correlation (0.714) with GII, suggesting that higher government revenue relative to GDP is also associated with higher gender inequality. These findings highlight the complex interplay between economic policies, government finances, and gender inequality in Sri Lanka. Moreover, the time series behaviour of each feature is presented in Fig 3.

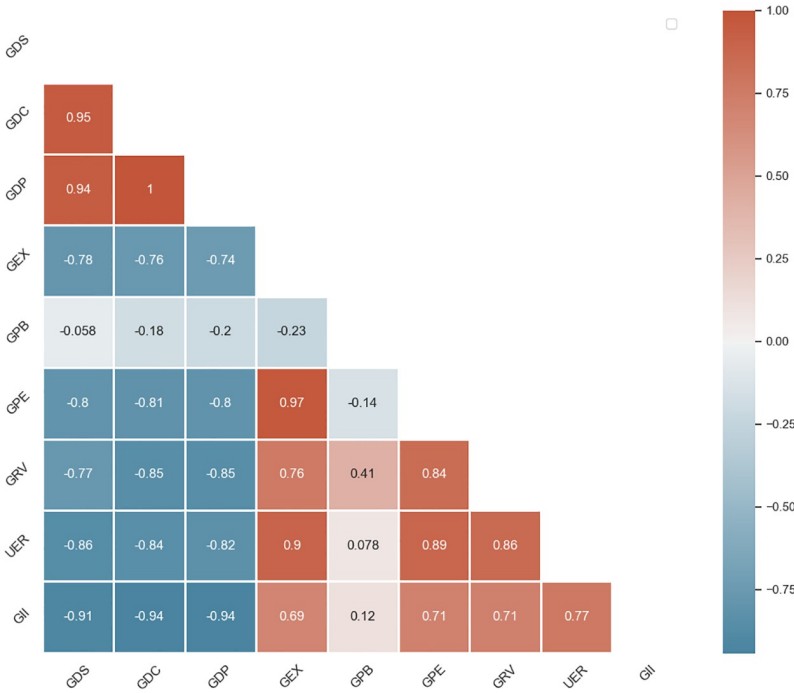

**Fig 2. Correlation matrix of key economic indicators and the GII.**

## Machine learning paradigms

The selection of ML models for the analysis was likely based on their suitability for the task of predicting the GII in Sri Lanka based on economic indicators.

1. Decision Trees (Fine and Medium): Decision trees are useful for this analysis because they can capture nonlinear relationships between the economic indicators and the GII. The "fine" and "medium" variants may represent different levels of complexity or pruning strategies applied to the decision tree to optimize its performance [58].

2. Ensemble Bagging: Ensemble methods like bagging can improve prediction accuracy by training multiple models on different subsets of the data and combining their predictions. This approach can reduce over-fitting and improve generalization to unseen data [59]. The prediction for bagged decision trees is the average prediction of all the trees in the ensemble (Eq 1):

$$\hat{y} = \frac{1}{N} \sum_{i=1}^{N} f_i(x) \tag{1}$$

where $\hat{y}$ is the predicted value, $N$ is the number of trees, and $f_i(x)$ is the prediction of the $i$-th tree.

3. Ensemble Boosting: Boosting algorithms, like AdaBoost or Gradient Boosting, iteratively improve the performance of weak learners (e.g., decision trees) by focusing on the instances that were misclassified in previous iterations. This can lead to higher predictive accuracy compared to individual models [60].
   The prediction for gradient boosting is the sum of predictions from multiple weak learners,

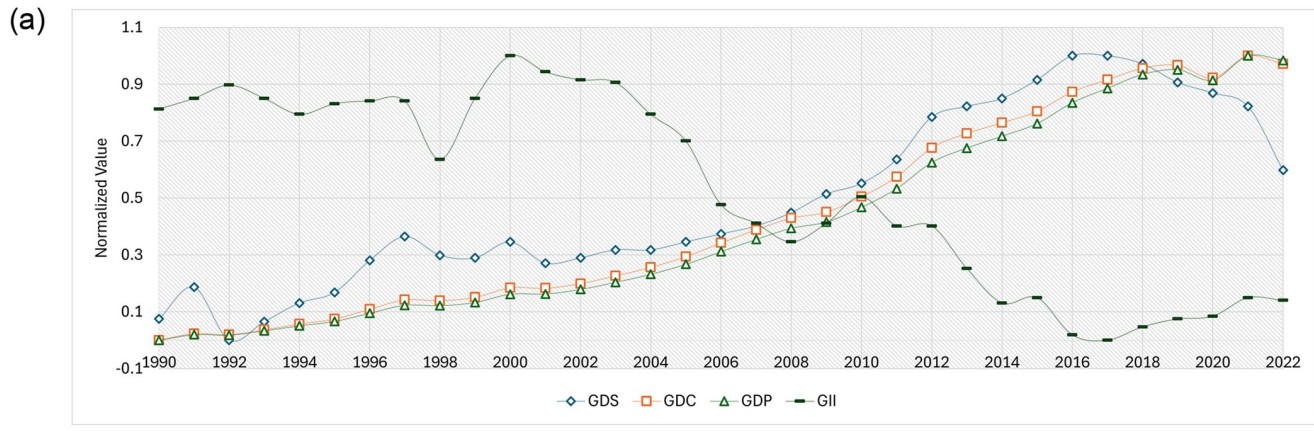

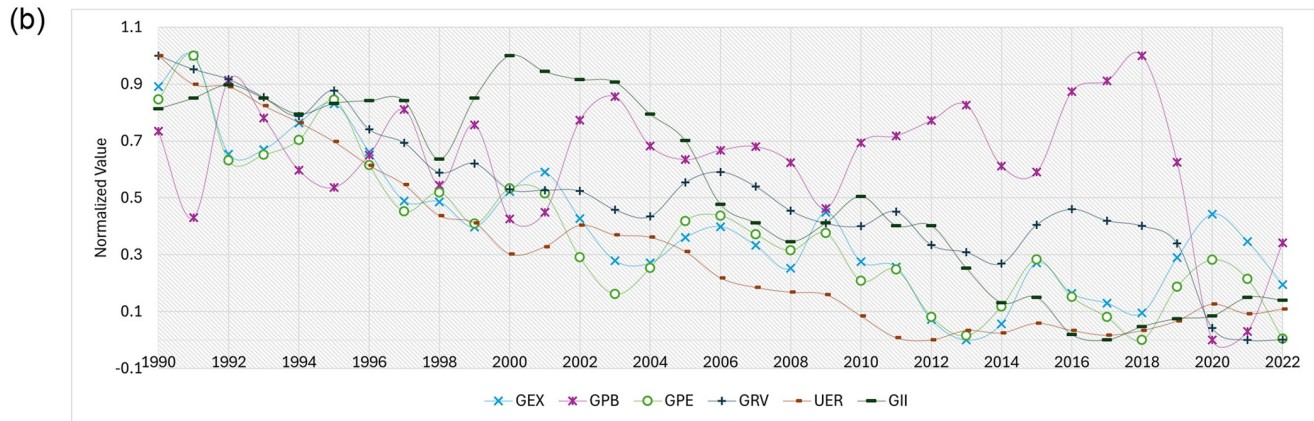

**Fig 3. Time series representation of the data.** (a) Features that have a Negative relationship with GII (b) Features that have a positive relationship with GII.

each correcting the errors of its predecessor (Eq 2):

$$\hat{y} = \frac{1}{N} \sum_{i=1}^{N} \alpha_i f_i(x) \qquad (2)$$

where $\hat{y}$ is the predicted value, $N$ is the number of weak learners, $\alpha_i$ is the learning rate for the $i$-th learner, and $f_i(x)$ is the prediction of the $i$-th learner.

4. Linear Regression: Linear regression is a simple yet effective model for analyzing the relationship between economic indicators and the GII. It assumes a linear relationship between the independent variables (economic indicators) and the dependent variable (GII) and can provide insights into the direction and strength of these relationships [61].
The prediction for linear regression is a linear combination of the input features (Eq 3):

$$\hat{y} = \beta_0 + \beta_1 x_1 + \beta_2 x_2 + \ldots + \beta_n x_n \qquad (3)$$

where $\hat{y}$ is the predicted value, $\beta$ is the coefficients, and $x$ are the input features.

5. Support Vector Machine (SVM) with Linear Kernel: SVMs are powerful for binary classification tasks like predicting gender inequality (high vs. low GII). The linear kernel is suitable

when the data is linearly separable, and SVMs are known for their ability to handle high-dimensional data [62].

The prediction for SVM is based on the distance of the input sample from the separating hyperplane (Eq 4):

$$\hat{y} = sign\left(\sum_{i=1}^{N_{sv}} \alpha_i y_i K(x_i, x) + b\right) \tag{4}$$

where $\hat{y}$ is the predicted class, $N_{sv}$ is the number of support vectors, $\alpha_i$ are the Lagrange multipliers, $y_i$ are the class labels, $x_i$ are the support vectors, $x$ is the input sample, $K$ is the kernel function, and $b$ is the bias term.

6. Gaussian Process Regression with Squared Exponential Kernel: Gaussian Process Regression (GPR) is a Bayesian approach that can capture complex nonlinear relationships in the data. The squared exponential kernel is commonly used for smooth functions and can model correlations between different features [63].

The prediction for GPR is a Gaussian distribution over the predicted values (Eq 5):

$$\hat{y} \sim N(\mu(x), \sigma^2(x)) \tag{5}$$

where $\hat{y}$ is the predicted value, $\mu(x)$ is the mean function, and $\sigma^2(x)$ is the variance function of the Gaussian process.

## Methodological framework from data acquisition to model evaluation

The methodological framework for this study involved several key steps, starting with data acquisition from the International Monetary Fund (IMF) [56] and World Bank data repositories [57]. The next step involved preprocessing the data, which included scaling numerical features.

After preprocessing, the data was analyzed to understand the relationships between the economic indicators and the GII. This involved calculating correlations between the variables to identify which indicators were most strongly associated with gender inequality. Based on these findings, the next step was to select the best ML algorithms for the dataset. This selection process likely involved comparing the performance of several algorithms, such as decision trees, ensemble methods like bagging and boosting, linear regression, support vector machines, and Gaussian process regression, using metrics like $R^2$, root mean squared error (RMSE), mean squared error (MSE), and mean absolute error (MAE).

Once the algorithms were selected, the models were trained on the dataset using the selected features and target variable (GII). The trained models were then tested on a separate portion of the dataset to evaluate their performance. This evaluation likely involved comparing the predicted values from the models to the actual values of the GII using the evaluation metrics mentioned earlier. The performance of each model was assessed, and the best-performing model(s) were selected for further analysis and interpretation of the results.

## Performance evaluation metrics

This section explores key performance metrics employed to assess the accuracy and reliability of the developed GII prediction models. The RMSE, MAE, and Coefficient of Determination ($R^2$) are discussed in detail, providing insights into the models' predictive capabilities. The analysis of these metrics enhances our understanding of the model's accuracy, highlighting their strengths and areas for improvement.

**Root mean squared error (RMSE).** The RMSE is a widely used metric that measures the average magnitude of the errors between predicted ($y_i$) and observed ($\bar{y}_i$) values (Eq 6) [47]. It comprehensively assesses the model's accuracy, where lower values indicate better performance [64].

$$RMSE = \sqrt{\frac{1}{N}\sum_{i=1}^{N}(y_i - \bar{y}_i)^2} \tag{6}$$

Where $N$ is the total number of observations.

**Mean absolute error (MAE).** MAE quantifies the average absolute difference between predicted ($\hat{y}_i$) and observed ($y_i$) values, providing a measure of the model's accuracy without considering the direction of the errors (Eq 7) [65].

$$MAE = \frac{1}{N}\sum_{i=1}^{N}|y_i - \hat{y}_i| \tag{7}$$

where N is the total number of observations.

**Coefficient of determination ($R^2$).** The Coefficient of Determination, often denoted as $R^2$, assesses the proportion of the variance in the dependent variable that is predictable from the independent variables (features). It ranges between 0 and 1, where higher values indicate better explanatory power (Eq 8) [46].

$$R^2 = 1 - \frac{\sum_{i=1}^{N}(y_i - \hat{y}_i)^2}{\sum_{i=1}^{N}(y_i - \bar{y}_i)^2} \tag{8}$$

Where $N$ is the total number of observations. $y_i$ is the observed value. $\hat{y}_i$ is the predicted value. $\bar{y}_i$ is the mean of the observed values.

These performance indexes collectively evaluate the model's predictive accuracy, highlighting different aspects of the prediction errors.

## Explainable AI using SHapley Additive exPlanations (SHAP)

When we use machine learning models to make predictions, such as predicting the Gender Inequality Index (GII) in Sri Lanka, it's important to understand why the model made certain predictions. SHAP is a technique that helps us do just that. It explains which factors (like GDP or government spending) are most important in influencing the model's decisions [66].

Imagine you're trying to decide whether to buy a car. You might consider factors like price, fuel efficiency, and brand. SHAP works like a checklist that shows which factors were most important in your final decision. In our study, SHAP helps us understand which economic indicators (like GDP or unemployment rate) are most crucial in predicting gender inequality.

SHAP is based on a concept from cooperative game theory, where it calculates the contribution of each feature (or economic indicator) to the prediction made by the machine learning model. Positive SHAP values indicate features that increase the predicted GII, while negative values indicate features that decrease the predicted GII. By analyzing these values, we can gain insights into the relative importance of different indicators in predicting gender inequality.

## Results and discussion

The results show the performance of various ML models in predicting the GII in Sri Lanka, as measured by the $R^2$ (Refer to Figs 4–7).

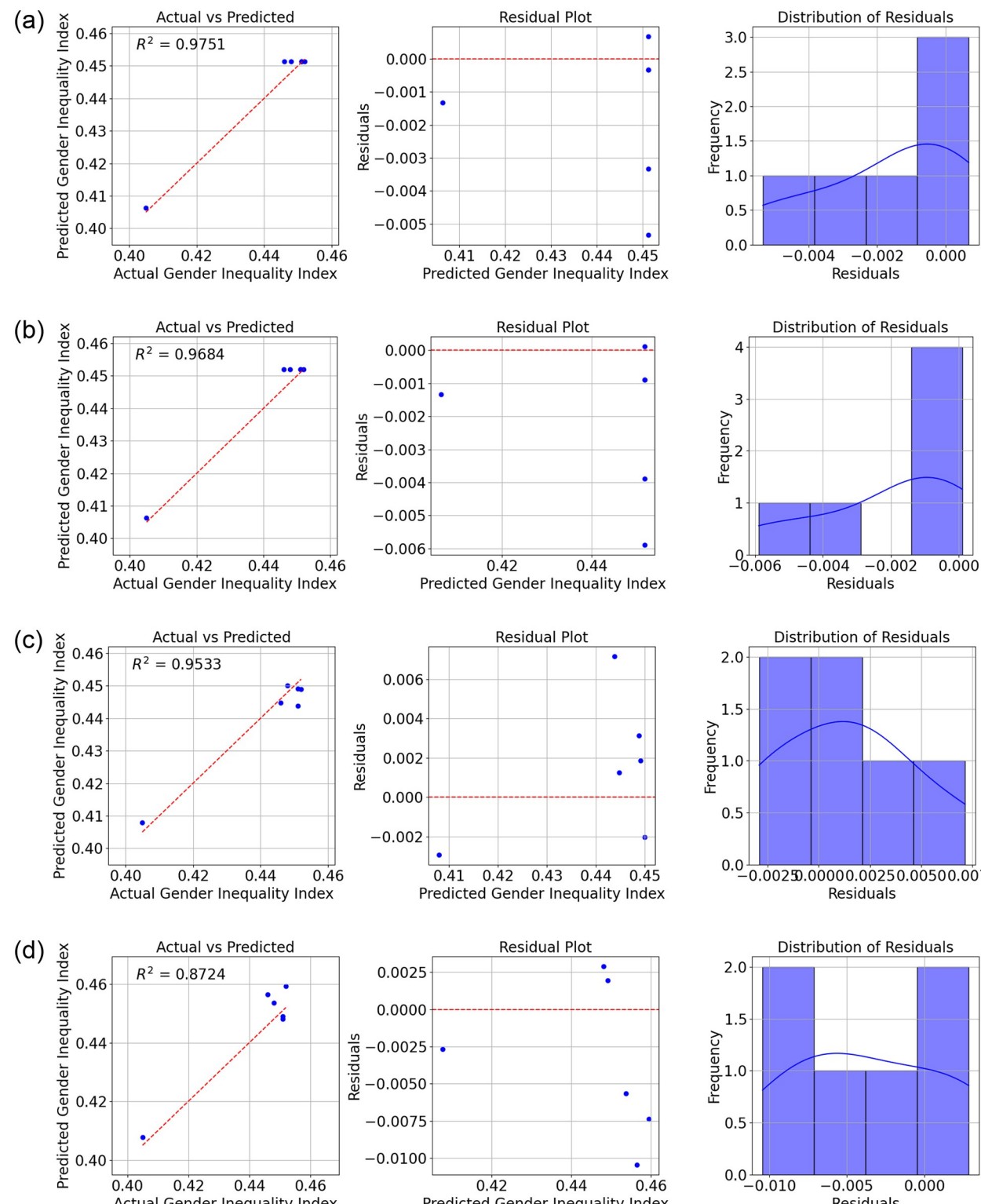

**Fig 4. Comparison of Actual and Predicted Values, Residual Plot, and Residual Distribution for (a) Fine Tree Regression (b) Medium Tree Regression (c) Ensemble Bagging and (d) Ensemble Boosting.**

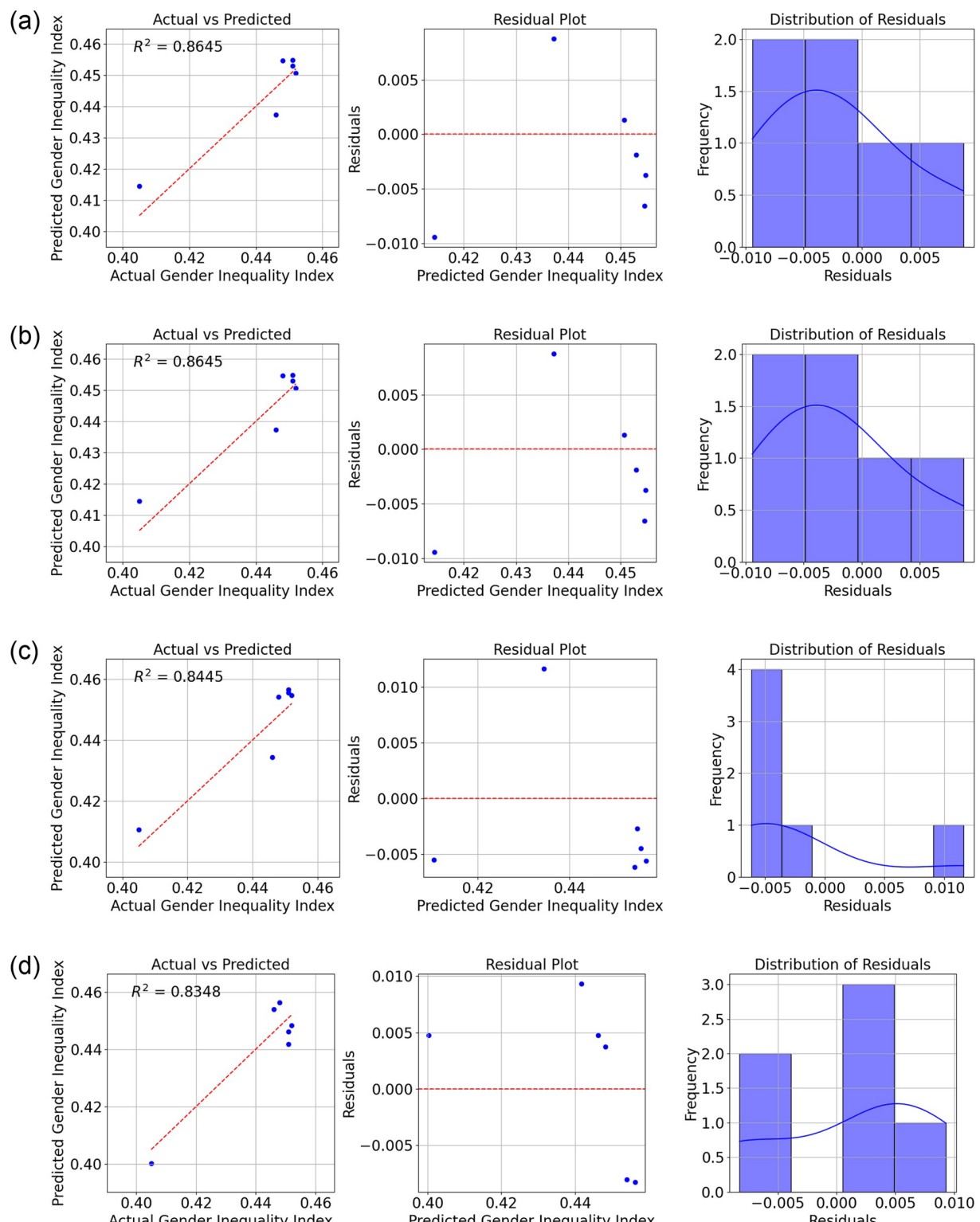

**Fig 5. Comparison of Actual and Predicted Values, Residual Plot, and Residual Distribution for (a) Linear Regression (b) Linear Regression—Robust (c) SVM and (d) GPR—Squared Exponential.**

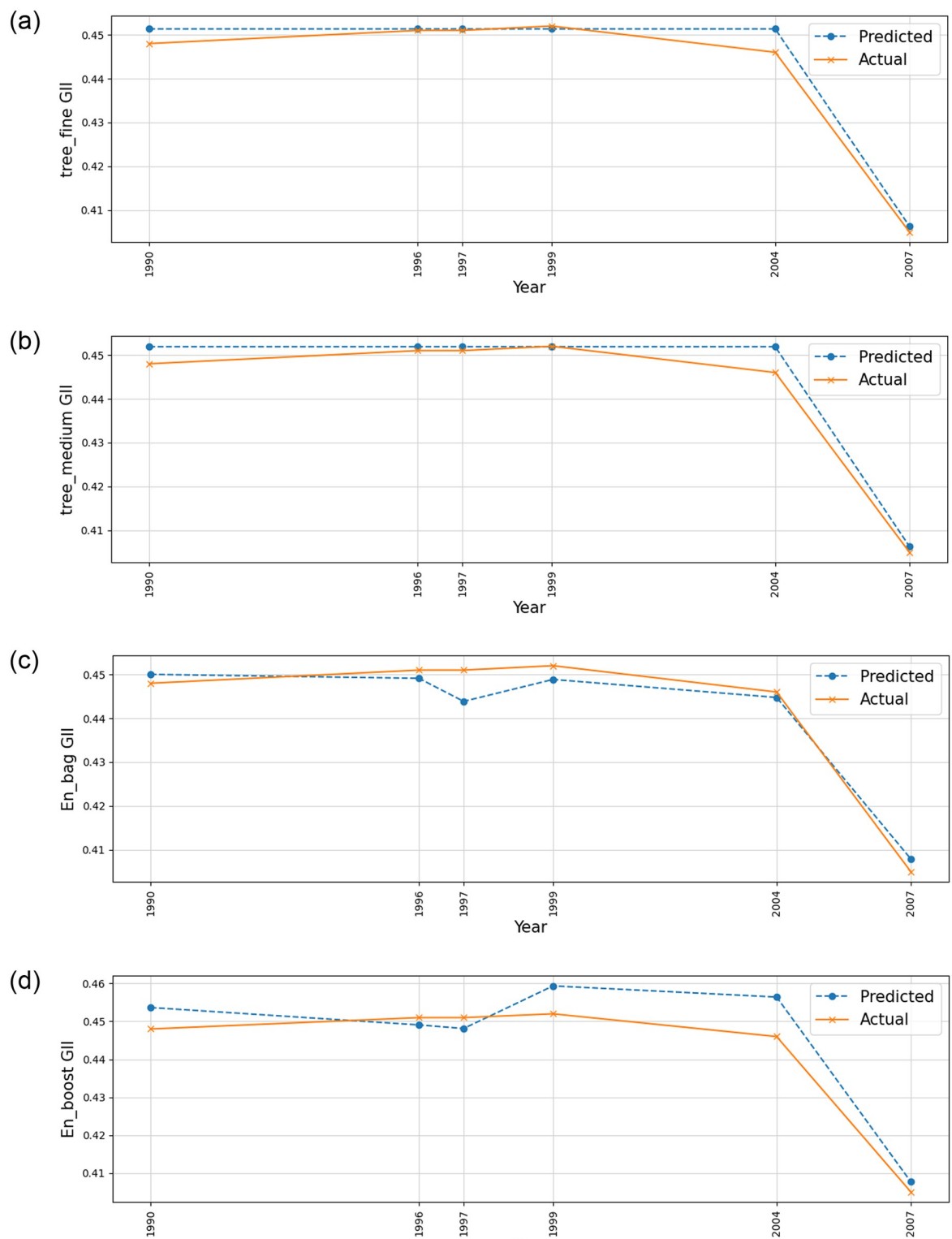

**Fig 6. Response Plot Showing Predicted and Actual Values Over Time for (a) Fine Tree Regression (b) Medium Tree Regression (c) Ensemble Bagging and (d) Ensemble Boosting.**

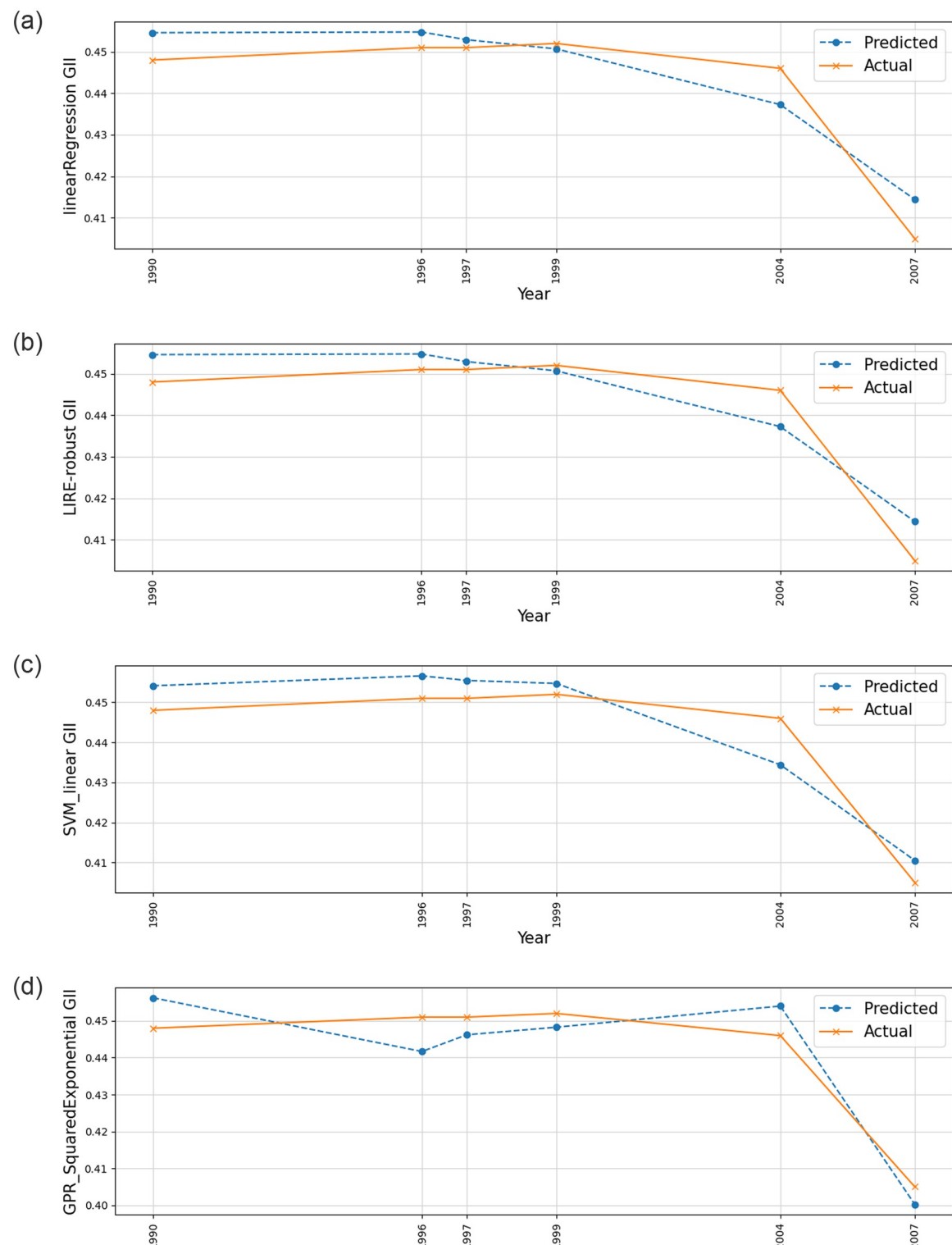

**Fig 7. Response Plot Showing Predicted and Actual Values Over Time for (a) Linear Regression (b) Linear Regression—Robust (c) SVM and (d) GPR.**

**Table 3. Performance metrics of machine learning models for predicting the GII in Sri Lanka.** The table shows the coefficient of determination ($R^2$), root mean squared error (RMSE), and mean absolute error (MAE) for each model.

| Model | $R^2$ | RMSE | MAE |
|---|---|---|---|
| Fine Trees | 0.9751 | 0.0026 | 0.0019 |
| Medium Trees | 0.9684 | 0.0030 | 0.0022 |
| Ensemble Bagging | 0.9533 | 0.0036 | 0.0031 |
| Ensemble Boosting | 0.8724 | 0.0060 | 0.0052 |
| Linear Regression | 0.8645 | 0.0062 | 0.0053 |
| Support Vector Machines | 0.8645 | 0.0062 | 0.0053 |
| Support Vector Machines (Linear) | 0.8445 | 0.0066 | 0.0060 |
| Gaussian Process Regression | 0.8348 | 0.0068 | 0.0065 |

From the data, we can see that the Fine tree regression model has the highest $R^2$ value of 0.975, indicating that it explains 97.5% of the variance in the GII. This suggests that the decision tree model with fine-tuning or pruning is highly effective in predicting gender inequality based on the selected economic indicators. The medium tree regression model also performs well, with an $R^2$ value of 0.968.

The ensemble bagging model and ensemble boosting model also show strong performance, with $R^2$ values of 0.953 and 0.872, respectively. These ensemble methods combine multiple models to improve prediction accuracy.

Linear regression and linear regression—Robust models achieved similar $R^2$ values of 0.864, indicating moderate predictive power. The SVM with a linear kernel performs slightly worse, with an $R^2$ value of 0.844.

Finally, Gaussian process regression with a squared exponential kernel has the lowest $R^2$ value of 0.835, suggesting that this model may not be the best choice for predicting gender inequality in this context.

Overall, the data highlights the importance of selecting the right ML model for the task, as different models can vary significantly in their predictive performance.

Table 3 shows the overall results of the study. The root mean squared error, mean squared error, and mean absolute error are important metrics for evaluating the performance of machine learning models in predicting the GII in Sri Lanka.

The RMSE measures the average magnitude of the errors in the predicted GII values compared to the actual values. A lower RMSE indicates that the model's predictions are closer to the actual values. The Fine Trees model achieved the lowest RMSE of 0.0026, followed closely by the Medium Trees model with an RMSE of 0.0029. The higher RMSE values for models like Ensemble Boost (0.0059) and GPR (0.0068) suggest that these models have larger errors in their predictions.

The MAE measures the average magnitude of the errors in the predicted GII values without considering their direction. A lower MAE indicates that the model's predictions are closer to the actual values. The Fine Trees model again performs the best with the lowest MAE of 0.0018, followed by Medium trees with 0.0021. Models like SVM (0.0060) and GPR (0.0064) have higher MAE values, suggesting larger errors in their predictions.

## Explanation of the feature importance in ML model training

SHAP values provide insights into how each feature contributes to an ML model's output. In this context, the SHAP values indicate the impact of different economic indicators on the prediction of the GII in Sri Lanka.

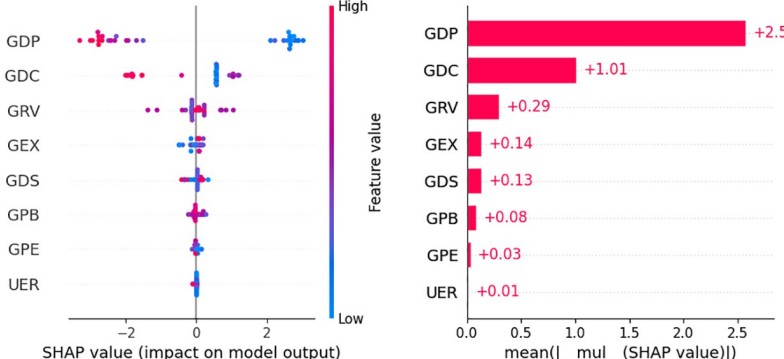

**Fig 8. SHAP values for key economic indicators in predicting the Gender Inequality Index (GII) in Sri Lanka.** On the right side, SHAP mean values and on the left side, Impact on model output.

Fig 8 shows the SHAP values for key economic indicators in predicting the GII in Sri Lanka. GDP has the highest impact, with a SHAP value of +2.57, suggesting that GDP is the most significant factor in predicting gender inequality. Similarly, GDC (GDP per capita) also plays an important role, with a SHAP value of +1.01. Other factors, such as government revenue (GRV) and government expenditure (GEX), have a smaller but still noticeable impact on the model's predictions.

These SHAP values are not just numbers; they are the key to understanding the model's decision-making process. They provide invaluable insights, helping us interpret the model's results and identify influential factors affecting gender inequality in Sri Lanka. Most importantly, they can inform policy decisions aimed at reducing gender disparities, empowering policymakers with crucial information.

### Hypothesis testing and empirical analysis

In this study, the primary objective was to investigate the relationship between key economic indicators and GII in Sri Lanka. To achieve this, we formulated hypotheses that certain economic factors—such as GDP, government expenditure, and unemployment rate—would have significant predictive power in explaining variations in GII.

**Testing of hypotheses using ML models.** The hypotheses were tested using multiple ML models, including Decision Trees, Ensemble Methods, linear regression, SVM, and Gaussian Process Regression. These models were chosen for their ability to capture complex relationships between the independent variables (economic indicators) and the dependent variable (GII).

The results of the ML models provided strong evidence supporting our hypotheses. For instance, the Fine Tree regression model, with an $R^2$ value of 0.975, indicated that the selected economic indicators collectively explain 97.5% of the variance in GII. This high level of explained variance suggests that the economic indicators we hypothesized as significant predictors indeed play a crucial role in determining gender inequality in Sri Lanka.

**Validation of hypotheses with SHAP analysis.** We utilized SHAP values to validate the results further and understand the contribution of each economic indicator to the GII.

The SHAP analysis revealed that GDP and GDC were among the most significant predictors of GII, with SHAP values of +2.57 and +1.01, respectively. These findings are consistent with our hypothesis that economic growth and development are inversely related to gender

inequality. The positive SHAP values indicate that increases in GDP and GDC contribute significantly to reducing GII, thus supporting the idea that economic prosperity is linked to greater gender equality.

On the other hand, GEX and UER also showed noticeable impacts on GII, albeit with different directions of influence. GEX had a positive SHAP value, suggesting that higher government spending might exacerbate gender inequality if not properly targeted, while UER's positive SHAP value indicates that higher unemployment rates are associated with higher GII, aligning with our hypothesis.

**Hypothesis testing.**   The combination of ML model results and SHAP analysis provides robust support for our initial hypotheses. The empirical analysis demonstrates that the economic indicators selected are not only correlated with gender inequality but also play significant roles in shaping it. By thoroughly testing these hypotheses through advanced statistical and machine learning techniques, our study contributes valuable insights into the socioeconomic factors that influence gender inequality in Sri Lanka.

## Gender inequality and policy implications in Sri Lanka

When examining the general pattern of the GII in Sri Lanka from 1990 to the present, it becomes clear that there have been notable fluctuations despite consistently low index values in recent years. A significant drop occurred in 1998, primarily due to a change in measurement methodology rather than an actual improvement in gender equality. Starting from 1998, the scope of the labor force participation rate among women was broadened to include unpaid female family workers, such as housewives, within the labor force which caused the sudden drop in the GII value [67]. In 2000, the GII reached its highest level since 1990, indicating an escalation in gender inequality. This increase was partly attributed to a rise in the maternal mortality ratio, which is one of the factors considered in calculating the GII.

Despite Sri Lanka achieving a GII score of 0.376 in 2022, ranking 90th and categorized as a country with high human development, a closer examination of the GII dimensions reveals areas for improvement. Specifically, Sri Lanka still faces challenges in enhancing women's labor force participation and increasing the representation of women in parliament. These persistent gender gaps highlight the need for continued efforts to promote gender equality in Sri Lanka [68].

The findings of this research suggest that macroeconomic factors such as GDS, GPC, GDP, GEX, GPB, GPE, GRV, and UER play a significant role in influencing the GII in the Sri Lankan context. This sheds light on areas that policymakers should prioritize in their efforts to enhance gender equality. In terms of labour force participation (age 15 and above), the gender gap was recorded at 41% in 2022, indicating a significant disparity that hinders gender equality [69]. Moreover, a significant proportion of economically active women are primarily engaged in unpaid family labor, notably in the agricultural sector [70]. An unexpected finding is that despite the higher enrollment of women in education in Sri Lanka, female unemployment rates consistently remain significantly higher, almost double that of men. This indicates the need for policy restructuring to address this issue. Occupational segregation, income disparities, employment discrimination, variations in job quality versus quantity for women, and limited entrepreneurial opportunities are among the demand-side constraints contributing to low female labor force participation [71].

Another significant factor exacerbating gender inequality in Sri Lanka is the under-representation of women in decision-making positions. For example, women's presence in both politics and higher managerial roles is strikingly minimal [72, 73]. "Glass ceilings" and "sticky floors," terms denoting barriers that prevent qualified individuals from advancing in their

careers due to discrimination, and biases that keep individuals in lower-level positions based on gender, are common obstacles contributing to the under-representation of women in decision-making roles.

When examining policy efforts targeting gender inequality in Sri Lanka, the National Policy on Gender Equality and Women's Empowerment outlines a framework for policy commitments spanning a decade, from 2023 to 2033 [74]. This framework encompasses a broad spectrum, including Economic Empowerment and Productive Employment, Social Equality and Empowerment, Equality in Decision-making, and more.

Sri Lanka has made significant strides in addressing gender disparities through various policy initiatives over the years, yet challenges persist alongside notable successes. On the positive side, programs like the National Framework for Women-Headed Households and the WIFI Suhuruliya Programme have provided crucial support to vulnerable groups, particularly women-led households, by offering financial assistance, training, and social services [75]. Additionally, policies such as the National Plan of Action on Sex and Gender-Based Violence and the National Action Plan for the Implementation of the UN Security Council's Resolutions on Women, Peace, and Security signify a commitment to combatting gender-based violence and promoting women's participation in peacebuilding efforts [76]. Nevertheless, despite these endeavors, progress is hindered by gaps in implementation, inadequate funding, and cultural obstacles. For instance, although laws may be in place to safeguard women's rights, weak enforcement mechanisms result in minimal real-world impact [77]. Additionally, deeply ingrained social norms and stereotypes frequently obstruct women's opportunities in education, employment, and leadership positions [78]. Another significant challenge encountered in these policy initiatives is the irregular implementation and sustainability of such measures. Political instability within the country often leads to the interruption of programs when there are changes in government [79]. Overcoming these challenges and attaining substantive gender equality requires persistent dedication, enhanced collaboration among stakeholders, and specific interventions targeting structural inequalities.

## Conclusion

In conclusion, this study provides a holistic analysis of gender inequality in Sri Lanka, focusing on the intricate interplay of various socioeconomic factors and their impact on GII using ML techniques. By analyzing a comprehensive dataset spanning from 1990 to 2022, including key economic indicators and the GII, this research sheds light on the nuanced dynamics of gender disparities in the country.

Strong relationships between economic indices like GDP, GEX, GPE, GRV, UER, and the GII have been found in this study. The direction of causality is still unknown despite the fact that these correlations show a clear association between gender inequality and economic determinants. It is equally possible that economic issues drive gender inequality or that a third, unmeasured component drives both. Consequently, more investigation is required to determine the causal mechanisms driving these associations.

As a developing nation classified within the lower-middle-income bracket by the World Bank, Sri Lanka demonstrates a negative correlation between the GII and factors like GDS, GDC, and GDP. This suggests the critical role of economic development in addressing gender disparities within the country. Moreover, indicators such as GEX, GPE, and GRV exhibit strong correlations with the GII, underlining the significance of the government's fiscal stance. Sri Lanka finds itself notably vulnerable in this regard, grappling with a persistent fiscal deficit.

ML models, including Decision Trees, Ensemble Bagging, Ensemble Boosting, Linear Regression, Support Vector Machine, and Gaussian Process Regression, were employed to

predict the GII based on economic indicators. The results demonstrate the effectiveness of these models in predicting gender inequality, with Decision Trees and Ensemble methods performing particularly well.

Furthermore, the application of explainable AI techniques, such as SHAP, helps us see which economic indicators are most important in predicting gender inequality. In our analysis, GDP and GDP per capita emerged as the key drivers.

Understanding these associations enables policymakers to examine which factors may influence gender inequality. While this study brings insight into the relationship between economic conditions and gender disparity, it should be noted that causality between these variables is yet to be suggested. Consequently, policy actions based on these findings should be handled with careful consideration. However, the high associations discovered specifically using AI, provide a solid platform for further investigation for future researchers, driving more targeted research and informed policy decisions in the future.

Overall, this research is not just a valuable addition to the existing literature on gender inequality. It goes beyond by providing a comprehensive analysis of socioeconomic factors and actionable insights into Sri Lanka's policy implications. Amidst the present economic crisis, Sri Lanka continues to demonstrate commendable levels of human development within the South Asian region. However, there remains potential for enhancement, particularly concerning gender disparity. The findings and insights generated from this study are not just informative but can also serve as a practical guide for evidence-based policy interventions aimed at reducing gender disparities and promoting gender equality in the country.

## Supporting information

**S1 Table. Description of variables in the model.**
(PDF)

## Author Contributions

**Conceptualization:** Sherin Kularathne, Amanda Perera, Namal Rathnayake.

**Data curation:** Sherin Kularathne, Namal Rathnayake.

**Formal analysis:** Sherin Kularathne, Amanda Perera.

**Funding acquisition:** Yukinobu Hoshino.

**Investigation:** Sherin Kularathne, Amanda Perera.

**Methodology:** Amanda Perera, Namal Rathnayake.

**Resources:** Amanda Perera.

**Software:** Sherin Kularathne.

**Supervision:** Upaka Rathnayake, Yukinobu Hoshino.

**Validation:** Upaka Rathnayake, Yukinobu Hoshino.

**Visualization:** Sherin Kularathne.

**Writing – original draft:** Sherin Kularathne, Amanda Perera, Namal Rathnayake.

**Writing – review & editing:** Upaka Rathnayake, Yukinobu Hoshino.

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
