## [Decision Letter · Decision Letter 0]

24 Jul 2024

PONE-D-24-24357Decoding Gender Inequality: A Holistic Analysis of Socioeconomic Factors and Policy Implications Using Machine LearningPLOS ONE

Dear Dr. Rathnayake,

Thank you for submitting your manuscript to PLOS ONE. After careful consideration, we feel that it has merit but does not fully meet PLOS ONE’s publication criteria as it currently stands. Therefore, we invite you to submit a revised version of the manuscript that addresses the points raised during the review process.

We look forward to receiving your revised manuscript.

Kind regards,

María del Carmen Valls Martínez, Ph.D.

Academic Editor

PLOS ONE

Journal Requirements:

This study was funded by the Japan Society for the Promotion of Science (JSPS) in the form of a Grants-in-Aid for Scientific Research (KAKENHI) grant to Prof Yukinobu Hoshino. Grant Number: 22KK0160.

4. Thank you for uploading your study's underlying data set. Unfortunately, the repository you have noted in your Data Availability statement does not qualify as an acceptable data repository according to PLOS's standards.

Reviewers' comments:

Reviewer's Responses to Questions

**Comments to the Author**

1. Is the manuscript technically sound, and do the data support the conclusions?

Reviewer #1: No

Reviewer #2: Yes

2. Has the statistical analysis been performed appropriately and rigorously? 

Reviewer #1: Yes

Reviewer #2: Yes

3. Have the authors made all data underlying the findings in their manuscript fully available?

Reviewer #1: Yes

Reviewer #2: Yes

4. Is the manuscript presented in an intelligible fashion and written in standard English?

Reviewer #1: Yes

Reviewer #2: Yes

5. Review Comments to the Author

**Reviewer #1: **This paper tries to interpret quite limited data in an apparently sparse field, and in doing so would add to the literature from which further, more nuanced research could expand. However, there are several small issues and one large one.

1. As mentioned in the introduction, the literature on the topic is thin. If that is the case, there is little that can be done for Sri Lankan women specifically, but I would expect to see more theoretical sociological literature included, or more general (non Sri Lankan) literature included. As it stands the literature review does little to support the paper's investigation or its findings.

2. The explanations of both rudimentary and complex statistical analysis methods is a nice idea, but should be clearer. Either write for an audience that already knows what SHAP is or for ones who don't know anything about statistics, otherwise both will be confused.

And finally,

3. The paper states multiple times that the findings show that economic factors play a significant and considerable role in shaping gender equality, but the causality is not investigated. You have shown correlations, but they may just as well show that gender equality shapes the economic factors. Without a stronger literature foundation or investigation of causality, this claim is unscientific and inappropriate. I suggest you review your findings and their implications closely, and either express more clearly something I have missed, or rephrase your conclusions to account for the unknown causality.

All that said, the idea of this paper is a fine one, you must only be aware the limitations of the methodology.

**Reviewer #2:** The paper explores the relationship between different socioeconomic factors and the gender inequality index (GII) in Sri Lanka. It uses different methodologies and compares them with each other to determine which one is the most appropriate, highlighting the use of machine learning (ML) techniques. The topic of the study is of great interest, is very topical and adds value to the existing literature. However, I believe that the paper needs to be restructured and needs to answer the research question more precisely. In the following paragraphs I detail my suggestions. I hope that the authors will find my comments useful in improving the paper.

Contribution: I consider the contribution of the paper to be adequate. The study addresses an interesting topic from different approaches. However, I think it lacks further theoretical development to support the research question. The content of the introduction is adequate, but the main hypothesis of the study should not be stated in the introduction and the variables should not be explained.

Literature review: despite being a fundamentally practical paper, the literature review section is scarce. I consider this to be the paper's biggest problem. There is a lack of a definition of the objective, a lack of support for the research with previous literature and a lack of specification of what is to be analysed in a concrete way. There is no theoretical underpinning throughout the paper to understand the relationships that are subsequently explained in the results. Hypotheses should be stated here. Moreover, there should be a separate hypothesis for each socioeconomic factor as the effect is different for each of them. On the other hand, the gender inequality index, which is the important variable in the study, is not justified, nor is it explained how it is calculated or its components.

Methodology and results: The methodology section is good. The procedure is clearly detailed, making it easy for the reader to understand the paper. As for the data and variables, the source of the data is explained in detail, but the detailed explanation of the variables is not as it was done in the introduction.

In the results section, the empirical analysis is very interesting and detailed. The graphs and tables facilitate the understanding of the results. The different results are much more detailed taking into account the different methodologies used than the effect of the different socioeconomic factors. At no point is a test of the previously stated hypothesis carried out.

Conclusions: I consider that the conclusions are correct. The results are correctly explained and the role of gender in this study, the causes and effects are argued for the first time in the document. The implications of the study in a field beyond the academic one are explained.

6. PLOS authors have the option to publish the peer review history of their article (what does this mean?). If published, this will include your full peer review and any attached files.

Reviewer #1: No

Reviewer #2: No

---

## [Author Response · Author response to Decision Letter 0]

27 Aug 2024

Please see the attached separate documents for Reviewer 1 and Reviewer 2 comments answers.

---

## [Decision Letter · Decision Letter 1]

11 Sep 2024

PONE-D-24-24357R1Decoding Gender Inequality: A Holistic Analysis of Socioeconomic Factors and Policy Implications Using Machine LearningPLOS ONE

Dear Dr. Rathnayake

Thank you for submitting your manuscript to PLOS ONE. After careful consideration, we feel that it has merit but does not fully meet PLOS ONE’s publication criteria as it currently stands. Therefore, we invite you to submit a revised version of the manuscript that addresses the points raised during the review process.

We look forward to receiving your revised manuscript.

Kind regards,

Vahid Mohamad Taghvaee

Academic Editor

PLOS ONE

Journal Requirements:

Additional Editor Comments:

The title should be edited to find a more specific form. It should specifically describe the current study rather than giving a general image of the literature.

Abstract:

Abstract should be edited. Instead of using “e.g.,” it can use other forms to more clearly express the results. In addition, it should specifically define the research implication, instead of offering a general policy intervention. Moreover, the methods should go to the method part of the abstract not in the results part.  Likewise is for the sample time period which should move to a more proper position.

Language:

Last but not the list, the manuscript requires a checking for language errors and academic writing style.

Reviewers' comments:

Reviewer 1#

Reviewer comments have been sufficiently addressed, the most important being the corrections to stated causailty, making it clear that strong correlations are not causal. The addition of further elaboration for SHAP values aligns nicely with the existing discussion of methods used, and the inclusion of literature review is appreciated. In future, I would recommend a more thorough connection between conclusions and literature, a connection to a theoretical framework can add a lot of substance to findings when correctly applied, but this is a matter of personal opinion.

Reviewer 2#

Thank you for reviewing the manuscript and addressing my comments in the first report. I believe that the document is now more complete, but I feel that there are still areas for improvement. I detail my suggestions in the following paragraphs. I hope that the authors will find my comments useful in improving the paper.

I still find the literature review lacking. The objective is now much clearer and the hypotheses are more specific, but theoretical support is still lacking. There is no mention of any theory being followed to formulate either the objective or the hypotheses. I reiterate the need to support the research with previous literature to substantiate the paper.

The changes made in the introduction do not seem to me to satisfy all the comments made in the first report, so I consider that it is necessary to restructure this section again (see the comments of the literature review of the first report).

Reviewer's Responses to Questions

**Comments to the Author**

1. If the authors have adequately addressed your comments raised in a previous round of review and you feel that this manuscript is now acceptable for publication, you may indicate that here to bypass the “Comments to the Author” section, enter your conflict of interest statement in the “Confidential to Editor” section, and submit your "Accept" recommendation.

Reviewer #1: All comments have been addressed

Reviewer #2: (No Response)

2. Is the manuscript technically sound, and do the data support the conclusions?

Reviewer #1: Yes

Reviewer #2: Yes

3. Has the statistical analysis been performed appropriately and rigorously? 

Reviewer #1: Yes

Reviewer #2: Yes

4. Have the authors made all data underlying the findings in their manuscript fully available?

Reviewer #1: Yes

Reviewer #2: Yes

5. Is the manuscript presented in an intelligible fashion and written in standard English?

Reviewer #1: Yes

Reviewer #2: Yes

6. Review Comments to the Author

Reviewer #1: Reviewer comments have been sufficiently addressed, the most important being the corrections to stated causailty, making it clear that strong correlations are not causal. The addition of further elaboration for SHAP values aligns nicely with the existing discussion of methods used, and the inclusion of literature review is appreciated. In future, I would recommend a more thorough connection between conclusions and literature, a connection to a theoretical framework can add a lot of substance to findings when correctly applied, but this is a matter of personal opinion.

Reviewer #2: Thank you for reviewing the manuscript and addressing my comments in the first report. I believe that the document is now more complete, but I feel that there are still areas for improvement. I detail my suggestions in the following paragraphs. I hope that the authors will find my comments useful in improving the paper.

I still find the literature review lacking. The objective is now much clearer and the hypotheses are more specific, but theoretical support is still lacking. There is no mention of any theory being followed to formulate either the objective or the hypotheses. I reiterate the need to support the research with previous literature to substantiate the paper.

The changes made in the introduction do not seem to me to satisfy all the comments made in the first report, so I consider that it is necessary to restructure this section again (see the comments of the literature review of the first report).

7. PLOS authors have the option to publish the peer review history of their article (what does this mean?). If published, this will include your full peer review and any attached files.

Reviewer #1: No

Reviewer #2: No

---

## [Author Response · Author response to Decision Letter 1]

3 Oct 2024

Please see the attached files for Editor's comments ("GII___Response_to_Editor_s_Comments.pdf") and Reviewer 2 comments ("GII___Response_to_Reviewer_2_r2_Comments.pdf").

---

## [Editor Report · Decision Letter 2]

7 Oct 2024

Analyzing the Impact of Socioeconomic Indicators on Gender Inequality in Sri Lanka: A Machine Learning-Based Approach

PONE-D-24-24357R2

Dear Dr. Rathnayake,

We’re pleased to inform you that your manuscript has been judged scientifically suitable for publication and will be formally accepted for publication once it meets all outstanding technical requirements.

Kind regards,

Vahid Mohamad Taghvaee

Academic Editor

PLOS ONE
---

## [Editor Report · Acceptance letter]

10 Oct 2024

PONE-D-24-24357R2 

PLOS ONE

Dear Dr. Rathnayake, 

I'm pleased to inform you that your manuscript has been deemed suitable for publication in PLOS ONE. Congratulations! Your manuscript is now being handed over to our production team.

Kind regards, 

on behalf of

Dr. Vahid Mohamad Taghvaee 

Academic Editor

PLOS ONE